# Recrystallization and Micronization of p-Toluenesulfonamide Using the Rapid Expansion of Supercritical Solution (RESS) Process

**Tsung-Mao Yang [1,2], Chie-Shaan Su [1], Jin-Shuh Li [2,*] , Kai-Tai Lu [2] and Tsao-Fa Yeh [2]**

[1]  Department of Chemical Engineering and Biotechnology, National Taipei University of Technology, Taipei 86222, Taiwan
[2]  Department of Chemical and Materials Engineering, Chung Cheng Institute of Technology, National Defense University, Taoyuan 86330, Taiwan
*   Correspondence: lijinshuh@gmail.com; Tel.: +886-3-3891716; Fax: +886-3-3892494

**Abstract:** This study is focused on the micronization of p-toluenesulfonamide (p-TSA) using the rapid expansion of supercritical solution (RESS) process. Taguchi's experimental design method was applied to determine the optimum operating conditions. $L_9(3^4)$ orthogonal array with four control factors and three levels of each control factor was used to design nine experimental conditions. Four control factors were selected, including extraction temperature, extraction pressure, pre-expansion temperature, and post-expansion temperature. The particle size and morphology of the prepared samples were observed by scanning electron microscopy (SEM). In addition, Fourier transform infrared spectrometer (FTIR), X-ray diffraction (XRD), and differential scanning calorimetry (DSC) were employed to compare the differences between the raw and micronized p-TSA particles. The experimental and analytical results indicated that the extraction temperature was the most significant factor for the micronization of p-TSA in the RESS process, and the optimal operating conditions were at an extraction temperature of 50 °C, an extraction pressure of 220 MPa, a pre-expansion temperature of 220 °C, and a post-expansion temperature of 30 °C. The p-TSA particles were micronized from the original average size of 294.8 μm to the smallest average size of 1.1 μm at the optimal RESS process conditions. Furthermore, the physicochemical characteristics of p-TSA did not differ significantly before and after recrystallization.

**Keywords:** p-toluenesulfonamide; RESS process; recrystallization; micronization

## 1. Introduction

p-Toluenesulfonamide (p-TSA) is a nonvolatile chemical that exists in solid form as white flakes or crystalline powder and is stable in neutral, acidic, or alkaline conditions. It is widely used as an organic synthesis intermediate in the fields of pharmaceuticals, pesticides, dyes, pigments, resins, and other organic target compound preparations [1–4]. p-TSA (CAS No. 70-55-3) is slightly toxic to algae, but non-toxic to fish and daphnids, which implies that the environmental risk is presumably low [5]. It is also approved by the US Food and Drug Administration for being applied into adhesives as food packaging materials. In recent years, p-TSA has been used as an anticancer drug for a variety of cancers, including hepatocellular carcinoma, non-small cell lung cancer, and tongue squamous cell carcinoma. It shows efficient anti-tumor activity in clinical trials through a local injection [6–8]. In addition, because of its lipophilic nature, p-TSA can easily penetrate and evenly distribute within tumors.

In the study of drug and pharmaceutical production, the high dissolution rate of drug powders is an important parameter to help the pharmaceutical developers get more effective products. One of the proposed techniques is to reduce the particle size of drug powders. It has been proven that the

dissolution rate is a direct function of the total surface area of the particles, and is inversely related to particle size. Therefore, the dissolution rate can be enhanced by using micronized drugs. In addition, small drug particles are also required in administration forms, which require the drug in micro-size size due to geometric reasons in the organ to be targeted (e.g., drugs for pulmonary use). In the way of reducing the particle size, several techniques have been reported, and a typical technique for preparing micron-sized drugs is mechanical comminution (e.g., by crushing, grinding, and milling) of previously formed larger particles. Although this technique is widely used, it is not an ideal way to produce small particles because drug substance properties and surface properties are altered in a mainly uncontrolled manner [9–11]. Therefore, techniques for directly preparing drug powders with a required particle size are of interest.

In the past two decades, one of the successful methods for producing micro- or nano-sizes drug powders with narrow particle size distribution is supercritical fluid (SCF) technology [12–18]. SCF technologies for micronized particles include: rapid expansion of supercritical solutions (RESS), supercritical antisolvent process (SAS), gas antisolvent process (GAS), aerosol solvent extraction system (ASES), solution enhanced dispersion by supercritical fluids (SEDS), and particles from gas-saturated solutions (PGSS). Most of these applications make use of carbon dioxide ($CO_2$) as the supercritical fluid, because it has a relatively low critical temperature and a moderate critical pressure ($T_c$ = 31.1 °C and $P_c$ = 7.38 MPa). In addition, it is inexpensive, non-toxic, and nonflammable. Among the mentioned methods, RESS is a simple and cost-effective way for production of fine, pure, and controllable particles and is demonstrated to be capable of producing fine particles in micro- or nano-scales. Li et al. [19] used a dynamic flow method to measure the solubility of p-TSA in supercritical $CO_2$ at different temperatures and pressure conditions. The experimental results showed that the solubility ranged from a p-TSA mole fraction of $0.97 \times 10^{-5}$ at 308.15 K and 8.0 MPa to $5.12 \times 10^{-5}$ at 328.15 K and 21.0 MPa. Since $CO_2$ is nonpolar, the solid-gas equilibrium behavior may be governed mainly by the physical interactions between $CO_2$ and the p-TSA molecules. The effect of pressure on the solubility of p-TSA follows the expected trend that the solubility increases with increasing pressure in an isothermal process. However, the effect of temperature on the solubility of p-TSA is more complicated and is determined by the solute vapor pressure, the solvent density, and the intermolecular interactions in the fluid phase.

In this study, the RESS process was used to micronize the p-TSA particles. The RESS process involved the saturation of supercritical $CO_2$ with p-TSA, followed by depressurization of the solution by passing through a heated nozzle into a low pressure chamber. The rapid decompression of the supercritical $CO_2$ containing the p-TSA drove the nucleation and the particle formation. Taguchi's experimental design method was applied to determine the optimum operating conditions. The effects of four control factors, including extraction temperature, extraction pressure, pre-expansion temperature, and post-expansion temperature, were evaluated from the experimental results. The particle size and morphology of the prepared samples were observed by scanning electron microscopy (SEM). In addition, the Fourier transform infrared spectrometer (FTIR), X-ray diffraction (XRD), and differential scanning calorimetry (DSC), were employed to compare the differences between the samples before and after recrystallization.

## 2. Experiment

### 2.1. Materials

Raw p-toluenesulfonamide (p-TSA, $CH_3C_6H_4SO_2NH_2$) with a purity greater than 99.0% was purchased from Sigma-Aldrich Co. Its crystal morphology is shown in Figure 1, and the average particle size is about 294.8 μm. Carbon dioxide ($CO_2$) with a purity of 99.9% was purchased from Qing-Feng Gas Co. (Taiwan) and was used as the solvent in the RESS process. All other chemicals were of reagent grade and used without further purification.

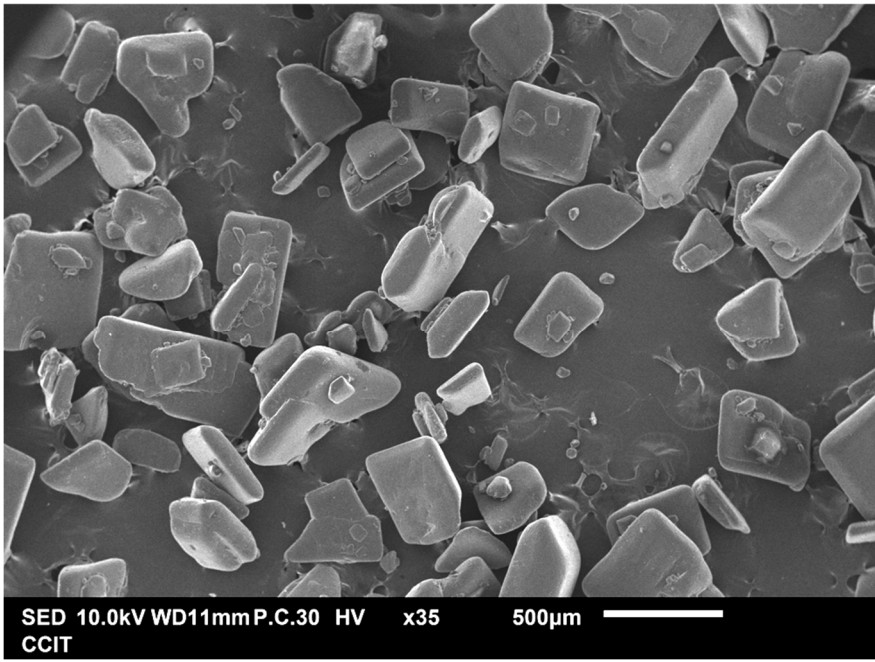

**Figure 1.** SEM image of raw p-TSA.

## 2.2. Apparatus and Procedure

The experimental apparatus used in this study for the RESS process was designed and made by Taiwan Supercritical Technology Co., Ltd. (Changhua, Taiwan) The schematic diagram of the RESS process is shown in Figure 2. The experimental system consisted of three main parts for the feeding of supercritical $CO_2$, the extraction of solute, and the particle formation. The pure $CO_2$ from the $CO_2$ cylinder (1) was first liquefied to 273.2 K by the cooler (2) and then compressed to the desired pressure by the high pressure pump (3). The system pressure was regulated and controlled by the back pressure regulator (A). The pressurized $CO_2$ was passed through the pre-heater (4) and then charged into the equilibrium cell (5) with the heating jacket (7). The system temperature was regulated and controlled by the temperature controller (6). The volume of the equilibrium cell was 150 $cm^3$. In each experiment, 15 g of the raw p-TSA was loaded into the equilibrium cell filled with 2 mm diameter glass beads. Two stainless filters were inserted at both ends of the equilibrium cell to avoid physical entrainment. Following p-TSA extraction with supercritical $CO_2$, the supercritical solution containing p-TSA was heated to a desired pre-expansion temperature by the heating tape (8) and subsequently expanded to the expansion vessel (10) through the capillary nozzle with an inner diameter of 100 μm (11) at the atmospheric pressure. The temperature of the expansion vessel was controlled at a desired post-expansion temperature by the water bath (12). The flow rate of expanded $CO_2$ was measured by the saturator (13) and the rotameter (14). The resulting p-TSA particles were finally collected in the expansion vessel for further analysis. Furthermore, the syringe (9) was used to inject 30–50 mL of ethanol into the sampling line to remove the residual p-TSA after the end of each experiment.

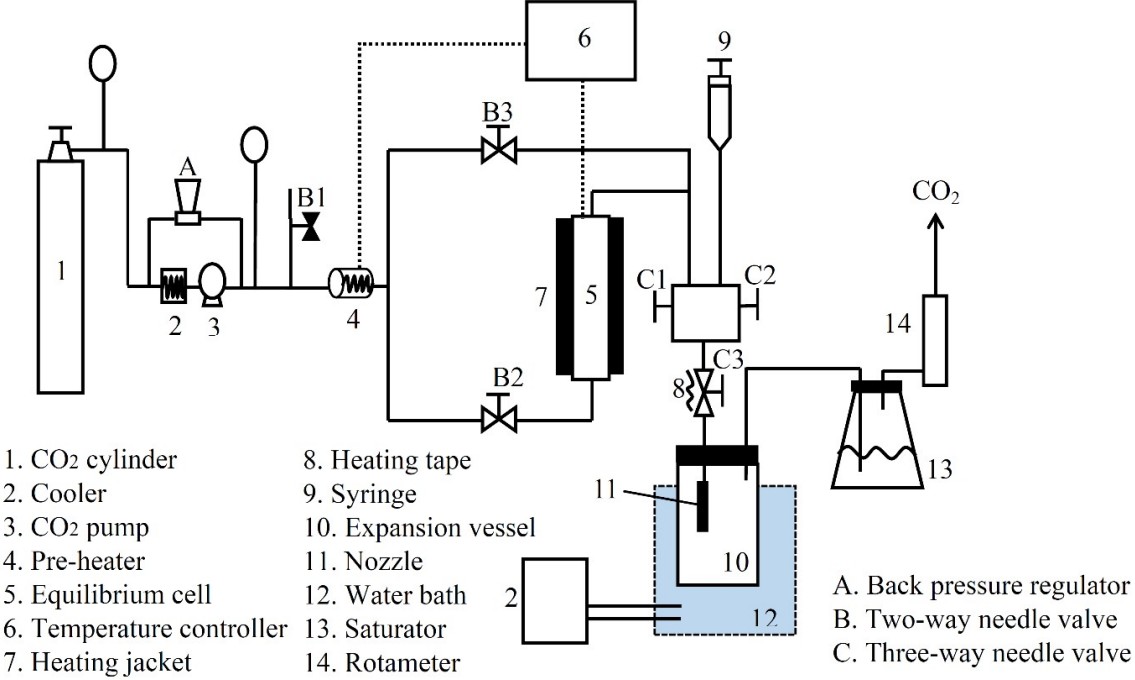

**Figure 2.** Schematic diagram of the experimental apparatus used for the RESS process.

The size and morphology of the micronized p-TSA particles were observed by scanning electron microscopy (SEM, Hitachi S-3000H, Tokyo, Japan). The particle size was determined using the Image J analysis software package (National Institutes of Health, Bethesda, MD, USA). The average particle size was calculated for at least 800 particles. In addition, Fourier transform infrared spectrometer (FTIR, PerkinElmer Spectrum 100, Waltham, MA, USA), X-ray diffraction (XRD, PANalytical X'Pert3 Powder, Malvern, UK), and differential scanning calorimetry (DSC, PerkinElmer DSC 4000, Waltham, MA, USA) were employed to compare the differences between the raw and micronized p-TSA particles. FTIR was used to analyze the sample in the 4000–500 $cm^{-1}$ region with a spectral resolution of 4 $cm^{-1}$. XRD was equipped with a Cu $K\alpha 1$ source. The sample was placed in the sample holder and was examined by scanning rate of 5° $min^{-1}$ in a 2θ range of 10–50° to analyze the phase composition and crystallinity of the sample. DSC measurements were carried out in an argon atmosphere at a heating rate of 10 °C $min^{-1}$ using sample weights in the range of 1–3 mg to study the effect of particle size on the thermal properties of p-TSA particles.

## 2.3. Design of Experiments

The Taguchi method is one of the most widely used design methods [20,21], which is making use of an orthogonal array from the experimental design to study more variables in lesser amount experiments. $L_9(3^4)$ orthogonal array with four control factors and three levels was referred to in the literatures [22,23], which was used to design the experiments and analyze the results in this study. Four control factors were selected, including extraction temperature, extraction pressure, pre-expansion temperature, and post-expansion temperature, each at three levels, as shown in Table 1. The above four factors were assigned to the $L_9(3^4)$ orthogonal array containing nine experimental conditions as shown in Table 2.

**Table 1.** The control factors and levels of the Taguchi experiments.

| Control Factor | Level | | |
|---|---|---|---|
| | 1 | 2 | 3 |
| A. Extraction temperature (°C) | 40 | 45 | 50 |
| B. Extraction pressure (MPa) | 180 | 200 | 220 |
| C. Pre-expansion temperature (°C) | 180 | 200 | 220 |
| D. Post-expansion temperature (°C) | 20 | 30 | 40 |

**Table 2.** The $L_9(3^4)$ orthogonal array of the Taguchi experiments.

| Exp. No. | Extraction Temperature (°C) | Extraction Pressure (MPa) | Pre-Expansion Temperature (°C) | Post-Expansion Temperature (°C) |
|---|---|---|---|---|
| A1 | 40 | 180 | 180 | 20 |
| A2 | 40 | 200 | 200 | 30 |
| A3 | 40 | 220 | 220 | 40 |
| A4 | 45 | 180 | 200 | 40 |
| A5 | 45 | 200 | 220 | 20 |
| A6 | 45 | 220 | 180 | 30 |
| A7 | 50 | 180 | 220 | 30 |
| A8 | 50 | 200 | 180 | 40 |
| A9 | 50 | 220 | 200 | 20 |

## 3. Results and Discussion

### 3.1. Analysis and Verification of Taguchi Experiments

Figure 3 shows the SEM images of micronized p-TSA particles under the experimental conditions A1–A9 of Taguchi orthogonal array, and Table 3 gives the average particle size under each experimental condition. Each experiment was run at least three times and the averaged values were obtained with a relative standard deviation of less than 5.3%. The experimental results were transformed into a signal-to-noise (S/N) ratio of using the Taguchi method. The S/N ratio was used as the measure of the effect of noise factors on the target characteristic. In general, there are three categories of the performance characteristics in the study of the S/N ratio: larger-the-better, smaller-the-better, and nominal-the-best. In this study, the average particle size of p-TSA was chosen as the quality characteristic. A smaller-the-better type of S/N ratio was used for analysis, and a smaller value indicated a better quality characteristic.

**Table 3.** Experimental results of Taguchi's orthogonal array.

| Exp. No. | Measured Particle Number (-) | Average Particle Size (μm) | Relative Standard Deviation (%) |
|---|---|---|---|
| A1 | 851 | 1.83 | 3.7 |
| A2 | 1088 | 1.39 | 4.0 |
| A3 | 1039 | 1.26 | 4.2 |
| A4 | 868 | 2.53 | 3.5 |
| A5 | 856 | 1.89 | 4.1 |
| A6 | 892 | 1.63 | 5.3 |
| A7 | 1285 | 1.45 | 4.1 |
| A8 | 922 | 1.37 | 4.7 |
| A9 | 927 | 1.40 | 4.6 |

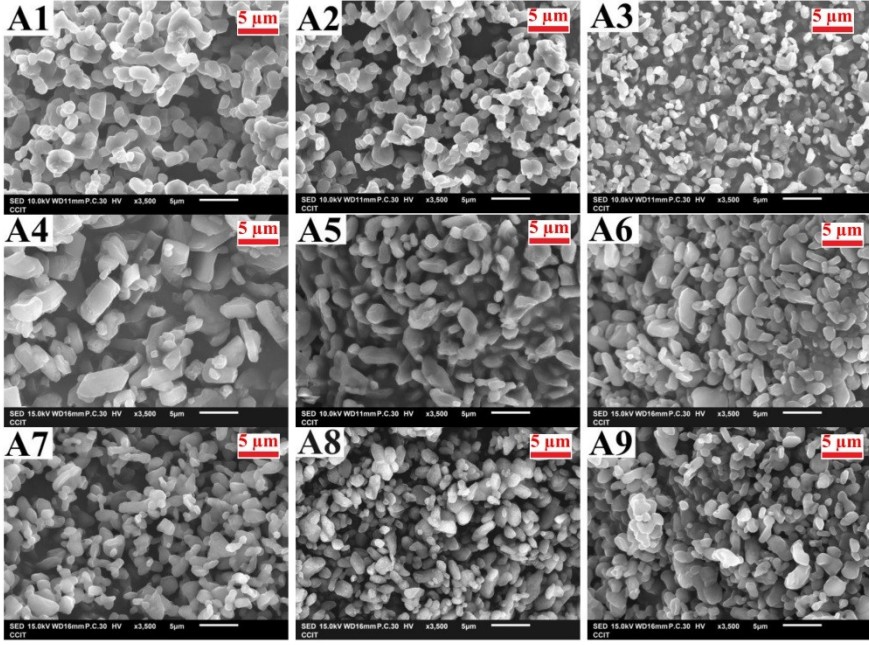

**Figure 3.** SEM images of micronized p-TSA particles under nine different experimental conditions.

Table 4 presents the range and contribution rank of each factor for the average particle size of p-TSA. In the table, the target values of different levels on each factor are the arithmetic average of target values corresponding to each level. The range is the difference between maximum and minimum of the target values on each factor and the rank represents the order of effect of each factor on this quality characteristic. The trend of parameter influence for four factors is shown in Figure 4. In the figure, the abscissa and ordinate represent the levels of factors and average particle size, and all points of corresponding levels for each factor are connected into a polyline, thus the change trend of the objective value for each factor with the increase of level can be obtained. Therefore, levels A3, B3, C3, and D2 have the smallest value of average particle size for the factors extraction temperature, extraction pressure, pre-expansion temperature, and post-expansion temperature, respectively. Based on above study, the minimum value of average particle size may be found on the condition that extraction temperature is 50 °C, extraction pressure is 220 MPa, pre-expansion temperature is 220 °C, and post-expansion temperature is 30 °C. It is also observed that the order of effect of each factor on this quality characteristic is extraction temperature > extraction pressure > pre-expansion temperature > post-expansion temperature. The extraction temperature (factor A) has the greatest influence.

**Table 4.** Range and contribution rank of each factor for average particle size (unit: μm).

| Level | Control Factors | | | |
| --- | --- | --- | --- | --- |
| | A (Extraction Temperature) | B (Extraction Pressure) | C (Pre-Expansion Temperature) | D (Post-Expansion Temperature) |
| 1 | 1.49 | 1.94 | 1.61 | 1.71 |
| 2 | 2.02 | 1.55 | 1.77 | 1.49 |
| 3 | 1.41 | 1.43 | 1.53 | 1.72 |
| Range | 0.61 | 0.51 | 0.24 | 0.23 |
| Rank | 1 | 2 | 3 | 4 |

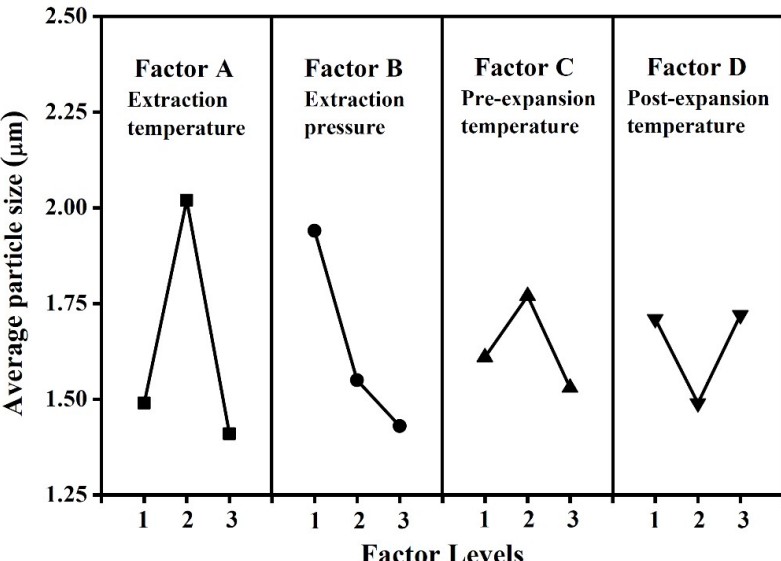

**Figure 4.** The trend of parameter influence for four factors on average particle size.

An additional confirmation experiment was performed to verify the effectiveness of the optimal parameters identified with the Taguchi method. The optimal design factor is the A3B3C3D2 parameter combination for small average particle size. Figure 5 shows the SEM image of micronized p-TSA particles under the experimental condition of the optimal parameter combination. The experiment was repeated five times under the same test condition and the average particle size was 1.1 μm with a relative standard deviation of 4.3%. The experimental results indicate that the A3B3C3D2 parameter combination produces a smaller average particle size than the other combinations tested in this study. The Taguchi method was successful in predicting the optimal parameter combination in order to achieve the minimum average particle size during the RESS process.

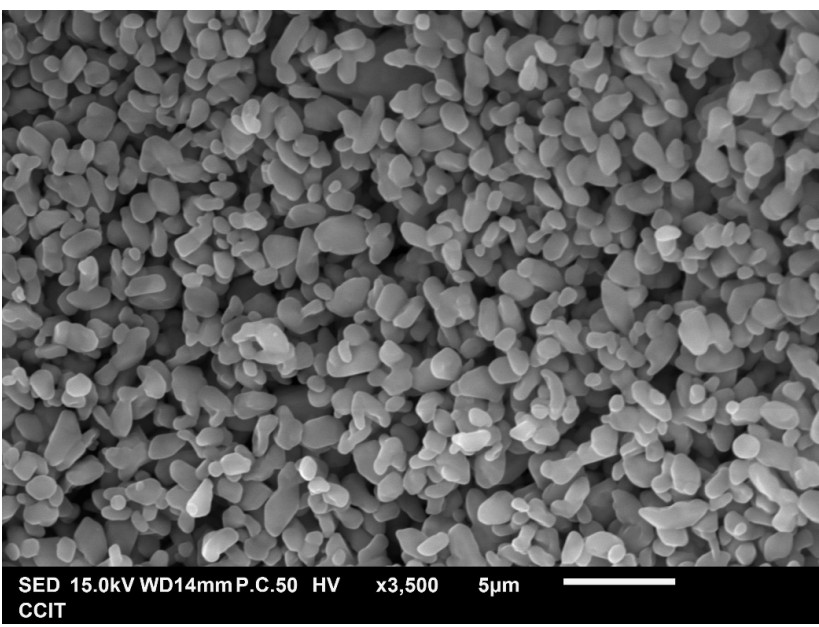

**Figure 5.** SEM image of micronized p-TSA particles under the experimental condition of the optimal parameter combination.

### 3.2. Identification of Micronized Products

The effect of the RESS process on the chemical structure of micronized p-TSA and its comparison with raw p-TSA were analyzed with FTIR. Figure 6 shows the FTIR spectra of raw and micronized p-TSA. The FTIR spectrum of micronized p-TSA displays similar absorption bands when compared with that of raw p-TSA. In these two spectra, aromatic C=C stretching is observed in 1475 and 1600 cm$^{-1}$, indicating the presence of a benzene ring structure. Ar-H and C-H stretchings are observed respectively in 3000 and 2900 cm$^{-1}$ which are part of the toluene structure. In addition, S=O stretching is found at 1475 and 1600 cm$^{-1}$, NH$_2$ stretching is found at 3350 and 3250 cm$^{-1}$, and N-H bending is found at 1560 cm$^{-1}$. These findings can confirm the presence of a sulfonamide functional group in the chemical structure. Based on the above FTIR analysis, it can be demonstrated that the RESS process does not affect the chemical structure of p-TSA.

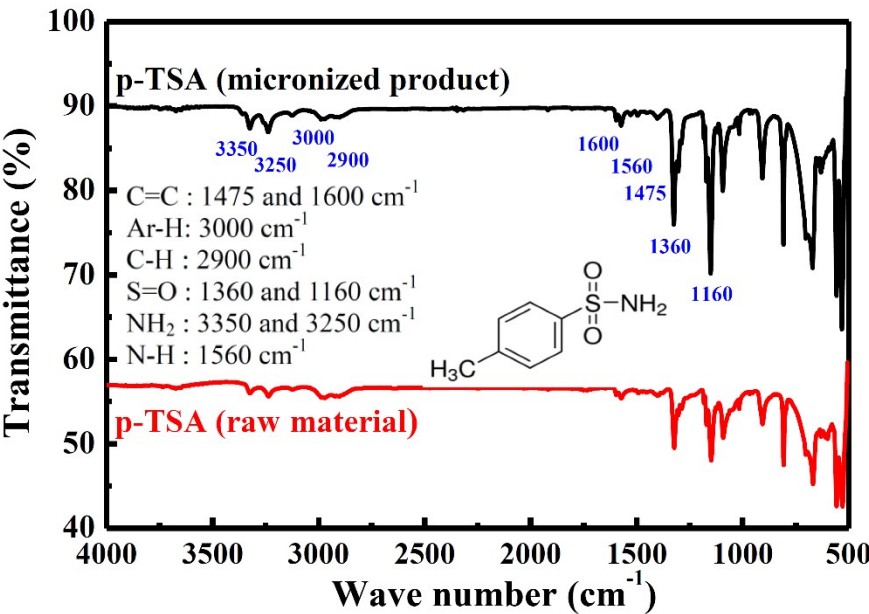

**Figure 6.** FTIR spectra of raw and micronized p-TSA.

Crystallinity affects material stability and physicochemical properties and can be studied by XRD analysis [24]. Figure 7 shows the XRD patterns of raw and micronized p-TSA. These two XRD patterns have almost the same diffraction peaks, but the XRD pattern of micronized p-TSA appears at the relatively lower peak intensity. Powell et al. [25] also reported a similar XRD pattern for p-TSA as shown in Figure 7. The degree of crystallinity usually decreases after the RESS process. Therefore, this reduction in peak intensity may be due to decreased crystallinity and particle size reduction. Keshavarz et al. [15] also reported similar experimental results for other materials.

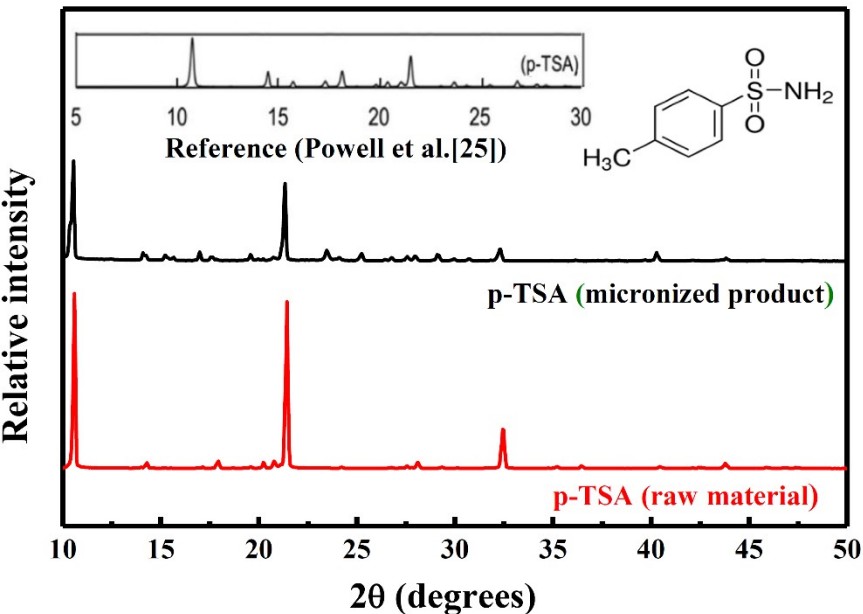

**Figure 7.** XRD patterns of raw and micronized p-TSA.

The DSC is a technique used to measure the energy and variation involved in phase change. It is also used to reflect the crystallinity and solid state stability of pharmaceutical compounds. Figure 8 shows the DSC thermograms of raw and micronized p-TSA, and these two curves reveal a similar endothermic peak at about 138 °C, which is consistent with the melting point of p-TSA. There is no significant change in the melting point, indicating that the solid states of particle crystals do not change after the RESS process. However, the melting point depression can be explained by the increase in thermal transfer surface and the decrease in degree of crystallinity. Furthermore, there is no significant change in the melting enthalpy.

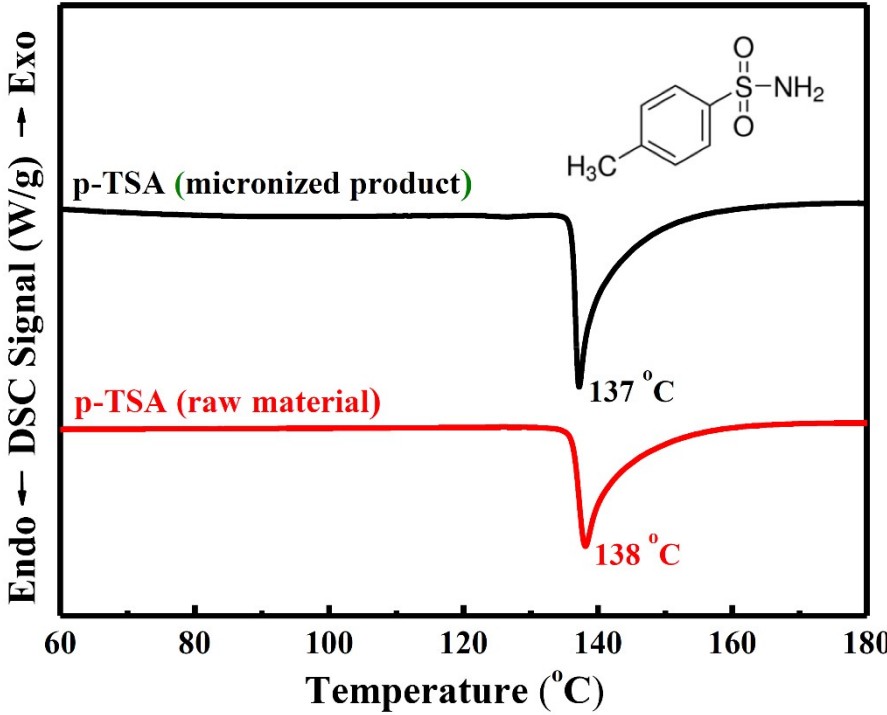

**Figure 8.** DSC thermograms of raw and micronized p-TSA.

## 4. Conclusions

In this study, micronization of p-TSA was successfully performed using the RESS process. Taguchi's experimental design method was applied to determine the optimum operating conditions for minimizing the average particle size. $L_9(3^4)$ orthogonal array with four control factors and three levels of each control factor was employed to analyze this quality characteristic. It was found and verified that the optimal operating conditions were at an extraction temperature of 50 °C, an extraction pressure of 220 MPa, a pre-expansion temperature of 220 °C, and a post-expansion temperature of 30 °C. The smallest average particle size of p-TSA obtained from the RESS processes was 1.1 μm, which had a significant size reduction from its original average particle size of 294.8 μm. In addition, it was also found that the extraction temperature was the most significant factor through the analysis of signal-to-noise (S/N) ratio. FTIR and XRD analyses showed that the chemical structure of p-TSA did not change significantly after the RESS process, and the crystallinity of p-TSA decreased. DSC analysis indicated that a slight decrease in the melting point of p-TSA after the RESS process can be explained by the increase in thermal transfer surface and the decrease in degree of crystallinity. This study suggests that the RESS process is a promising method for significantly reducing the particle size of pharmaceuticals in order to enhance their dissolution rate and bioavailability.

**Author Contributions:** Conceptualization, T.-M.Y. and C.-S.S.; methodology, J.-S.L., K.-T.L., and T.-F.Y.; formal analysis, T.-M.Y.; investigation, T.-M.Y.; experiment, T.-M.Y.; data curation, T.-M.Y.; writing-original draft preparation, T.-M.Y., C.-S.S. and J.-S.L.; writing-review and editing, K.-T.L. and T.-F.Y; supervision, T.-F.Y.; project administration, C.-S.S.; funding acquisition, C.-S.S.

**Funding:** This research was financially supported by the Ministry of Science and Technology, R.O.C. under grant number of MOST 106-2221-E-027-113.

**Conflicts of Interest:** The authors declare no conflicts of interest.

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
