# Peer review of "Recrystallization and Micronization of p-Toluenesulfonamide Using the Rapid Expansion of Supercritical Solution (RESS) Process"

_crystals, doi:10.3390/cryst9090449_

Round 1

Reviewer 1 Report

This paper is well written. I would like to suggest some minor changes. The counts should be shown on the y axis of figure 7.

In describing the XRD results there are two sentences that need to be improved.

"These two XRD patterns almost have the same diffraction peaks, but the XRD pattern of micronized p-TSA appears the relative lower peaks intensity" needs to be rephrased and

"Therefore, this reduction in peaks intensity may be due to decreased crystallinity and particle size reduction." Remove the second s.

.

Reviewer 2 Report

This is a very interesting manuscript dealing with the particle size reduction of p-toluenesulfonamide using the rapid expansion of supercritical solution process. A classical Taguchi's experimental design method was applied to determine the optimum operating conditions for particle size reduction. Original average size of 294.8 μm was reduced to the smallest average size of 1.1 μm at the optimal process conditions. With comparison purposes, the physicochemical properties of original raw material and micronized particles were analyzed by scanning electron microscopy, Fourier transform infrared spectrometer, X-ray diffraction and differential scanning calorimetry. The manuscript could be accepted after checking some minor points as shown in the attached pdf file. In particular the following:

Some minor typo and grammar mistakes are observed as shown in the pdf file. Lines 67-69: Please check this sentence because this is clear for hte solubility of gases in liquids but p-TSA is solid. Thus, normally the solubility of solids in liquids is not affected in a significant way with pressure. Regarding Fig. 3: Please show more clearly the size scale. It is unreadable. Line 213: It could be interesting to compare the enthalpies of fusion if available.
